# Correct use and ease-of-use of placebo ELLIPTA dry-powder inhaler in adult patients with chronic obstructive pulmonary disease

**Thomas M. Siler**[1], **Renu Jain**[2]*, **Kathryn Collison**[2¤], **Raj Sharma**[3], **Laura Sutton**[2]*, **Jamie Rees**[4], **David I. Bernstein**[5]

1 Midwest Chest Consultants, St Charles, Missouri, United States of America, 2 US Medical Affairs, GSK, Research Triangle Park, Durham, North Carolina, United States of America, 3 Respiratory Medical Franchise, GSK, Brentford, United Kingdom, 4 Biostatistics, Respiratory, GSK, Brentford, United Kingdom, 5 Division of Immunology, Allergy and Rheumatology University of Cincinnati College of Medicine and Bernstein Clinical Research Center, Cincinnati, Ohio, United States of America

¤ Current address: AstraZeneca, Cambridge, United Kingdom
* renu.g.jain@gsk.com (RJ); laura.b.sutton@gmail.com (LS)

**Data Availability Statement:** All relevant data are within the paper and its Supporting Information files.

## Abstract

### Background

Inhaler technique errors are common in chronic obstructive pulmonary disease (COPD) treatment, potentially leading to poor disease management. Our pooled analysis approach assessed correct use and ease-of-use of a placebo ELLIPTA dry-powder inhaler (DPI) in patients with COPD.

### Methods

Adults with COPD from open-label/non-blinded studies evaluating a placebo ELLIPTA DPI and reporting outcomes of correct use (based on the ELLIPTA DPI patient information leaflet [PIL]) and/or ease-of-use were included. Correct use and ease-of use at study end were primary and secondary endpoints, respectively. Data from patients in the placebo ELLIPTA DPI arm of each study were pooled, and the intent-to-treat (ITT) population was used for all analyses.

### Results

Four placebo ELLIPTA DPI studies, reporting correct use (n = 4) and ease-of-use (n = 2), were included in the analysis. The ITT population comprised 1232 patients (mean age 66.2 years). For the primary endpoint, 80.1% (n = 975/1217) of patients demonstrated correct use at study end (95% confidence interval [CI]: 77.8%–82.3%). For the secondary endpoint, 95.7% (n = 797/833) of patients rated placebo ELLIPTA DPI use "easy"/"very easy" at study end (95% CI: 94.1%–97.0%). Correct use and "easy"/"very easy" user ratings remained high across younger (40–64 years) and older (≥65 years) age groups.

**Funding:** This work was funded by GSK (Study 212146). The funders had a role in study design, data collection and analysis, decision to publish, and preparation of the manuscript.

**Competing interests:** RJ, KC, RS, LS, and JR report employment with, and stock/share ownership in, GSK during study conduct. KC is currently employed by AstraZeneca and LS is no longer employed by GSK. TMS received research support from West-Ward Pharmaceuticals, Theravance Biopharma US, Inc., GSK, Pearl Therapeutics, Chiesi, AstraZeneca, Novartis, Boehringer Ingelheim, Forest, Compleware, Evidera, Oncocyte, Teva, Vapotherm, Sunovion, Proterix BioPharma, Seer, and Sanofi. TMS has also received speaker fees from GSK, Mylan Inc./ Theravance Biopharma US, Inc., and Sunovion, and consulting fees from Vapotherm. DIB received grant/research/clinical trial support from GSK, Teva, AstraZeneca, Pearl Therapeutics, Novartis, Genentech, Inc., Merck, Boehringer Ingelheim, Amgen, Aimmune, Shire, and Biocryst and consulted/participated in advisory boards for GSK, ALK America, Gerson-Lehman, and Guidepoint Global. This does not alter our adherence to PLOS ONE policies on sharing data and materials.

## Conclusions

Across age groups, most patients used the placebo ELLIPTA DPI correctly and rated it "easy"/"very easy" to use. Consistent with the Global Initiative for Chronic Obstructive Lung Disease 2021 report, our findings emphasize that proper training and clear instructions on PILs are important for optimal inhaler use.

## Introduction

Inhaled therapy is a mainstay in the management of chronic obstructive pulmonary disease (COPD) [1]. Pressurized metered dose inhalers (pMDIs) and dry-powder inhalers (DPIs) are the most commonly used delivery systems in COPD [2–4], with each inhaler having varying techniques for proper use [5]. Inhaler technique depends on proper inhaler preparation and handling before inhalation, patient training, and optimal inhalation pattern [6]. The Global Initiative for Chronic Obstructive Lung Disease (GOLD) 2022 report advises that inhaler technique should be assessed to direct prescription of appropriate therapy and correct poor inhaler technique prior to escalating therapy [1]. However, errors in inhaler technique are common [7–9] and are particularly prevalent in elderly patients [10, 11]. These errors can result in inadequate drug delivery to the lungs [6] and thus poor efficacy, resulting in poorer outcomes and suboptimal disease control [1, 2, 8, 12], underscoring the importance of developing inhalers that patients find easy to operate correctly.

ELLIPTA is a DPI developed for the delivery of inhaled medications [13]. It was revealed that patients with COPD of different severities can generate sufficient inspiratory flow (minimum >43 L/min) to ensure adequate drug delivery with the ELLIPTA inhaler, which requires flow rates of 30–90 L/min for a delivered dose [13–16]. Due to the known problems associated with incorrect inhaler technique [1, 2, 6, 8, 12], it is important to identify whether patients can correctly use the ELLIPTA inhaler and whether they find it easy to use. The aim of our study was to assess correct use and ease-of-use of a placebo ELLIPTA DPI in adult patients with COPD using a pooled analysis approach.

## Materials and methods

### Study design

The internal GSK study database was used to identify studies for inclusion in the analysis and their corresponding primary publication was identified. Open-label/non-blinded studies of adult patients with a diagnosis of COPD that evaluated a placebo ELLIPTA DPI, reported outcomes of correct use and/or ease-of-use, and reported a completion date by October 2018, were included. The inhalers used in the study contained placebo, allowing for a focus on inhaler attributes and removing the possibility of bias due to treatment effects. Studies not evaluating a placebo ELLIPTA DPI, review articles, and conference abstracts were excluded.

No institutional review board review and approval or informed consent procedures were required for this study as no new patients were recruited.

### Data source and variables

The following raw data on patients who used the placebo ELLIPTA DPI were extracted from each study included in the pooled analysis: characteristics of study participants (including age, sex, race, and duration of COPD), ease-of-use and correct use of the placebo ELLIPTA DPI,

and safety (adverse event [AE] and serious AE [SAE]) findings. The intent-to-treat (ITT) population comprised all patients who contributed to the final analysis in each study. The ITT population was used for all study population, outcomes, and safety analyses. The inhalers used in the study contained placebo, however, routine collection of AEs was conducted in line with regulatory guidelines for all clinical trials included in this analysis.

### Endpoints

The primary endpoint was the percentage of participants who demonstrated correct use of the placebo ELLIPTA DPI at study end, determined by healthcare professional (HCP) assessment of inhaler technique. Correct use was defined as the patient correctly completing the following steps based on the ELLIPTA inhaler patient information leaflet (PIL) [14]: cover of the inhaler was opened; inhaler was not shaken; complete exhalation before inhalation; exhalation away from the mouthpiece; correct inhalation maneuver (a series of steps including one long, steady, deep breath in through the mouth) [16]; lips sealed round mouthpiece; no obstruction of air inlet; breath held after medication inhaled; inhaler closed after use. The secondary endpoint was the percentage of participants rating the placebo ELLIPTA DPI as "easy" or "very easy" to use at study end, as assessed by questionnaires in patients deemed to be using the placebo ELLIPTA DPI correctly at study end.

### Synthesis of results

The data were collected and summarized for all individual studies. Datasets were pooled into one main dataset from which all analyses were conducted. For all study endpoints, 2-sided 95% confidence intervals (CIs) for percentages were calculated using the exact binomial distribution. The endpoint data were also analyzed by age subgroup (40–64, 65–74, and ≥75 years) using descriptive and inferential statistics related to CIs only. The heterogeneity of studies was not assessed.

## Results

### Study selection and characteristics

Four of 36 identified placebo ELLIPTA DPI studies were included in the analysis after meeting pre-defined eligibility criteria (Fig 1). Characteristics of each of the studies included in the analysis are reported in Table 1. All included studies (200301 [17], 201071 [18], 206215 [19], and 206901 [20]) evaluated correct use of the placebo ELLIPTA DPI and 2 of the 4 studies (200301 [17] and 201071 [18]) evaluated ease-of-use of the placebo ELLIPTA DPI. Three studies (200301 [17], 206215 [19] and 206901 [20]) were crossover studies using two different placebo inhalers in a randomized sequence; the placebo ELLIPTA DPI and a comparator placebo inhaler with which they had no recent experience. Study 201071 [18] used the placebo ELLIPTA DPI only. Of the 2 studies that used an ease-of-use questionnaire, study 200301 [17] used 1 version, while patients in study 201071 [18] were randomized to receive version A or B; the only difference between the versions was that response options were in an alternative order.

AE, adverse event; ATS, American Thoracic Society; COPD, chronic obstructive pulmonary disease; DPI, dry powder inhaler; ERS, European Respiratory Society; HCP, healthcare professional; ICS, inhaled corticosteroid; IFU, instructions for use; ITT, intent-to-treat; LABA, long-acting beta2-agonist; LAMA, long-acting muscarinic antagonist; MDI, metered dose inhaler; PIL, patient information leaflet; SABA, short-acting beta2-agonist; UK, United Kingdom; US, United States.

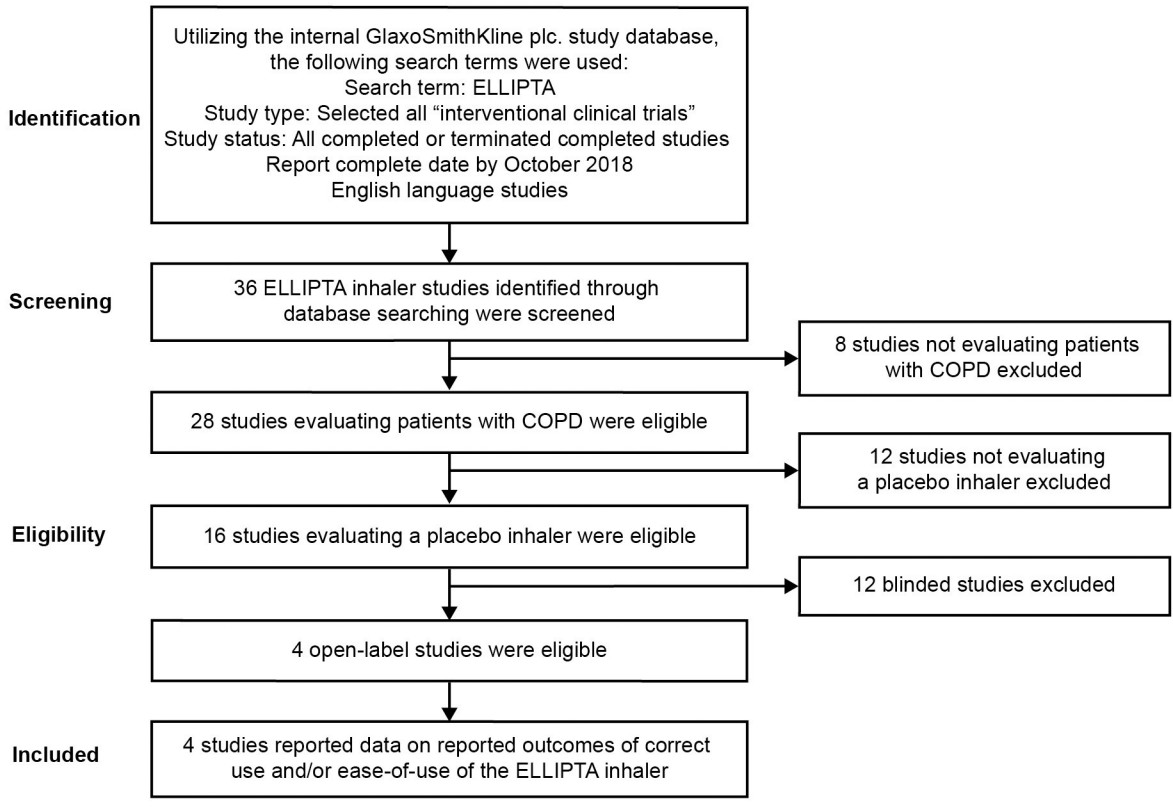

**Fig 1. Database search flow diagram. COPD, chronic obstructive pulmonary disease.**

**Table 1. Study characteristics.**

| Study publication (Study number/ Trial registration) | Study design[a] and location | Key patient eligibility criteria | ITT population, n | Device(s) assessed | Key outcomes assessed[b] | Device training schedule | Study duration, days |
|---|---|---|---|---|---|---|---|
| van der Palen et al. 2016 (200301/ NCT02184624) [15] | Randomized, cross-over, multicenter study at 2 centers in the UK and 6 centers in the Netherlands | **Inclusion criteria** • Aged ≥40 years and with a primary diagnosis of COPD, as defined by ATS/ERS criteria [21] • Naïve to ELLIPTA DPI use and ≥1 other inhaler device **Exclusion criteria** • History of allergy/ hypersensitivity to lactose/milk protein or magnesium stearate or to any other excipient found in commercially available inhaler devices • Current diagnosis of asthma | 567 | The following devices were assessed vs. ELLIPTA DPI in 5 separate substudies: • DISKUS • MDI • Turbuhaler • HandiHalerBreezhaler | • Correct use, assessed by a trained HCP using a checklist of inhaler errors (based on the ELLIPTA DPI PIL) after patients had read the PIL (primary endpoint[c]) [14] • Ease-of-use, assessed using an ease-of-use questionnaire (see S1 Table) • AEs | Patients were asked to read the PIL of the device and then asked to perform inhaler use. Errors were recorded by a trained respiratory nurse. If the patient made any errors, the nurse demonstrated correct use of the inhaler and the patient was asked to perform inhaler use again. If the patient continued to make errors, the nurse demonstrated the process again up to a maximum of 3 times. | 1 |

*(Continued)*

**Table 1.** (Continued)

| Study publication (Study number/ Trial registration) | Study design[a] and location | Key patient eligibility criteria | ITT population, n | Device(s) assessed | Key outcomes assessed[b] | Device training schedule | Study duration, days |
|---|---|---|---|---|---|---|---|
| Feldman et al. 2019 (201071/ NCT02586493) [16] | Randomized, single-arm, multicenter study at 17 centers in the US | **Inclusion criteria** • Aged ≥40 years, with an established diagnosis of COPD as defined by ATS/ERS criteria [21], and receiving maintenance inhaler therapy for COPD **Exclusion criteria** • Current diagnosis of asthma Use of the DPI within the previous 6 months | 267 | Placebo ELLIPTA DPI | • Correct use, assessed by a trained HCP using a correct use checklist (based on the ELLIPTA DPI PIL) on Visit 2 (Day 28) after a single attempt without further instruction (patients read the PIL before Visit 1 only) [14] • Ease-of-use, assessed using an ease-of-use questionnaire (primary endpoint) (see S2 Table) • AEs | Comprised 2 study visits and a telephone call. At Visit 1 (Day 1), the screening procedure involved an assessment of correct inhaler use within 3 attempts. Prior to their first attempt, subjects reviewed written instructions for the correct use of the inhaler based on the PIL but did not receive any training. Subjects were permitted to receive training (verbal instruction and a demonstration of correct use) from the HCP in between attempts 1 and 2, and/or attempts 2 and 3, if necessary. Subjects unable to demonstrate the correct use of the placebo inhaler within 3 attempts at Visit 1 were considered screening failures and did not continue in the study. | 28 |
| van der Palen et al. 2018 (206215/ NCT02982187) [17] | Randomized, cross-over, multicenter study at 2 centers in the UK and 3 centers in the Netherlands | **Inclusion criteria** • Aged ≥40 years and current or former smokers with ≥10 pack-years of smoking history, with a documented COPD history, as defined by ATS/ERS criteria [21] • Receiving maintenance therapy with a fixed-dose ICS/ LABA combination inhaler either with or without concurrent LAMA therapy during the preceding 4 weeks. Short-acting rescue medications were permitted **Exclusion criteria** • Current diagnosis of asthma and patients with recent experience (within 2 years) of the ELLIPTA DPI, any capsule inhaler, the DISKUS inhaler, or the Turbuhaler | 159 | The following devices were assessed vs. ELLIPTA DPI in 2 separate substudies: • DISKUS + HandiHalerTurbuhaler + HandiHaler | • Correct use, assessed by a trained HCP using a checklist of inhaler errors (based on the ELLIPTA DPI PIL) after patients had read the PIL (primary endpointc) [14] • AEs | For each device, patients were asked to read the PIL and were then observed by an HCP for correct use. If there were errors, the HCP provided instructions on correcting the observed errors and the patient attempted correct inhaler use again. The process could be repeated once more if the second attempt was unsuccessful, but no more than 3 attempts were permitted. | 1 |

*(Continued)*

**Table 1.** (Continued)

| Study publication (Study number/ Trial registration) | Study design[a] and location | Key patient eligibility criteria | ITT population, n | Device(s) assessed | Key outcomes assessed[b] | Device training schedule | Study duration, days |
|---|---|---|---|---|---|---|---|
| Kerwin et al. 2020 (206901/ NCT03227445) [18] | Randomized, cross-over, multicenter study at 17 centers in the US | **Inclusion criteria** Aged ≥40 years and current or former smokers with >10 pack-years of smoking history, with a documented COPD history, as defined by ATS/ERS criteria [21] **Exclusion criteria** Diagnosis of asthma >1 COPD exacerbation requiring hospitalization in the 12 months prior to randomization Use of the ELLIPTA DPI, DISKUS, or HandiHaler DPI in the 12 months prior to randomization Receipt of current COPD maintenance treatment by any of the study inhalers Receipt of a SABA only for COPD maintenance | 239 | ELLIPTA DPI DISKUS + HandiHaler | Correct use, defined as the patient making 0 errors from an error checklist based on the ELLIPTA DPI PIL (primary endpoint)[14] AEs | At randomization, participants were asked to read the IFU section of the approved prescribing information, and then demonstrated use of the assigned inhaler(s) to an HCP. In the case of errors, the HCP gave the participant verbal advice to ensure correct technique was understood before leaving the clinic. The participant took the inhaler(s) and IFU home for 28 days (Period 1). On returning to the clinic at Visit 2 (Day 28), participants were assessed for correct use of their first assigned inhaler(s) and were then given their next assigned inhaler(s) and IFU for Period 2. Demonstration of correct inhaler use was again required, and HCPs advised if the participant made any errors before taking the appropriate inhaler(s) and IFU home. Participants returned on Visit 3 (Day 56) for correct use of the second assigned inhaler, assessed by the HCP. | 56 (only 28 days on ELLIPTA DPI) |

[a]Study designs did not meet the eligibility criteria (open-label/non-blinded study design).

[b]Correct use and ease-of-use data also analyzed by age subgroups (40–64, 65–74, and ≥75 years).

[c]Primary endpoint defined as proportion of patients making ≥1 critical inhaler error after reading the PIL.

In all 4 studies, HCPs assessing the patients were trained in the correct use of each inhaler using error checklists created primarily using the respective PILs. The non-ELLIPTA checklists differed slightly between studies based on local-level inhaler instructions. The ELLIPTA inhaler checklists were identical within all studies [17–20]. One study (200301 [17]) included patients with COPD who were naïve to the ELLIPTA inhaler, while 3 studies included patients who were required to have had no use of the ELLIPTA inhaler within the 6 months (201071 [18]), 12 months (206901 [20]), or 24 months (206215 [19]) prior to screening (Table 1).

## Patients

The 4 studies that met the inclusion criteria screened a total of 1265 patients. Of these, 1232 patients were included in the ITT population, and 1217 completed their respective studies (Fig 2). Patient demographics and baseline characteristics of the ITT population are summarized in Table 2. Patients' mean age was 66.2 years and 31% had a COPD history of ≥10 years.

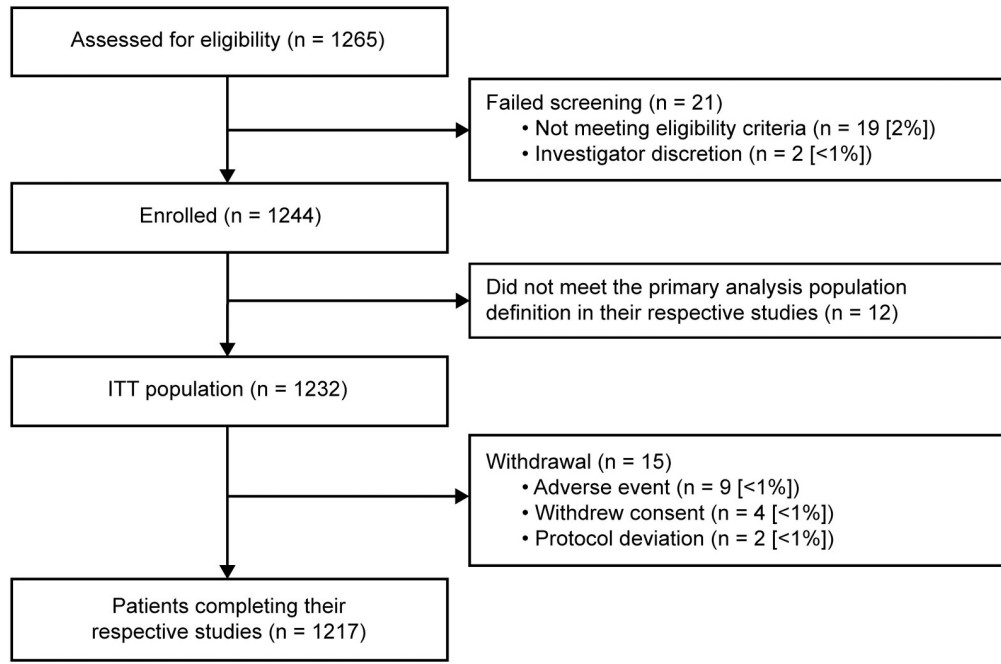

**Fig 2. Patient flow, including study completion and withdrawal. ITT, intent-to-treat.**

**Table 2. Baseline patient demographics and clinical characteristics.**

| Characteristic | ITT population (N = 1232) |
|---|---|
| Mean age, years (SD) | 66.2 (8.45) |
| Age group, n (%) | |
| 40–64 years | 485 (39) |
| 65–74 years | 556 (45) |
| ≥75 years | 191 (16) |
| Female, n (%) | 531 (43) |
| Race, n (%)[a] | |
| White | 1134 (92) |
| African American or African | 52 (4) |
| Asian | 43 (3) |
| Other | 2 (<1) |
| Duration of COPD, n (%) | |
| <1 year | 72 (6) |
| ≥1 to <5 years | 377 (31) |
| ≥5 to <10 years | 402 (33) |
| ≥10 to <15 years | 227 (18) |
| ≥15 years | 154 (13) |

[a]Data was missing for 1 patient.

Percentages do not equal 100 due to rounding.

COPD, chronic obstructive pulmonary disease; ITT, intent-to-treat; SD, standard deviation.

**Table 3. Efficacy outcomes for individual studies.**

| Study publication (Study number/Trial registration) | Primary endpoint: correct use of inhaler | Secondary endpoint: ease-of-use of Inhaler | Overall (critical and non-critical) errors using inhaler |
|---|---|---|---|
| van der Palen et al. 2016 (200301/ NCT02184624) [15] | After reading the PIL only, fewer patients had ≥1 critical error using the placebo ELLIPTA DPI compared with DISKUS (5% vs. 44%), MDI (13% vs. 60%), Turbuhaler (8% vs. 44%), HandiHaler (14% vs. 48%), or Breezhaler (13% vs. 46%) (all p<0.001). After reading the PIL, the majority of patients made no errors using ELLIPTA DPI (57–70% across the 5 substudies), thus not requiring instruction from the nurse | A larger proportion of patients in each substudy rated ELLIPTA DPI "very easy" or "easy" to use compared with DISKUS (97% vs. 60%), MDI (92% vs. 44%), Turbuhaler (96% vs. 55%), HandiHaler (98% vs. 38%), or Breezhaler (94% vs. 55%) (all p<0.001) | After reading the PIL only, fewer patients with COPD had ≥1 overall error using ELLIPTA DPI compared with DISKUS (30% vs. 65%), MDI (31% vs. 85%), Turbuhaler (31% vs. 71%), HandiHaler (43% vs. 62%), or Breezhaler (31% vs. 56%) (all p<0.001) |
| Feldman et al. 2019 (201071/ NCT02586493) [16] | In the COPD study, 258 (97%) patients demonstrated correct use of the placebo ELLIPTA DPI at the end of the study | Among subjects demonstrating the correct use of ELLIPTA DPI at study end, 93% of COPD subjects rated the inhaler as "easy" or "very easy" to use | In the COPD study, 8 (3%) patients made an error using ELLIPTA DPI at the end of the study |
| van der Palen et al. 2018 (206215/ NCT02982187) [17] | In both substudies, significantly fewer patients using the placebo ELLIPTA DPI made ≥1 critical error after reading the PIL compared with patients using DISKUS + HandiHaler or Turbuhaler + HandiHaler. In each substudy, 9% (n = 7/80) of patients made ≥1 critical error using the placebo ELLIPTA DPI vs. 75% (n = 60/80) using DISKUS + HandiHaler in substudy 1 (p<0.001) and 73% (n = 58/79) of patients using Turbuhaler + HandiHaler in substudy 2 (p<0.001) | Not analyzed | Significantly fewer patients using the placebo ELLIPTA DPI had ≥1 overall error in the first attempt after reading the PIL, compared with patients using other inhalers (24% vs. 80% with DISKUS + HandiHaler and 22% vs. 80% with Turbuhaler + HandiHaler) (both p<0.001) |
| Kerwin et al. 2020 (206901/ NCT03227445) [18] | The number of patients demonstrating errors (critical and non-critical errors) was 9 (4%) for the placebo ELLIPTA DPI, 14 (6%) for DISKUS, and 27 (12%) for HandiHaler in the ITT population (n = 239) at Day 28 | Not analyzed | Per primary endpoint column data |

COPD, chronic obstructive pulmonary disease; DPI, dry-powder inhaler; ITT, intent-to-treat; MDI, metered dose inhaler; PIL, patient information leaflet.

## Endpoints

For the primary endpoint, 80.1% (n = 975/1217) of patients demonstrated correct use of the placebo ELLIPTA DPI at study end (95% CI: 77.8%–82.3%), assessed at a 1-day visit for studies 200301 [17] and 206215 [19], and Day 28 for studies 201071 [18] and 206901 [20]. Notably, at study end, the study that enrolled patients who were naïve to the ELLIPTA inhaler (200301 [17]) showed an error rate of 30–43%, compared with an error rate of 3–24% among the 3 studies [18–20] allowing prior but not recent use of the ELLIPTA inhaler (Table 3). For the secondary endpoint, 95.7% (n = 797/833) of patients using the placebo ELLIPTA DPI rated it as "easy" or "very easy" to use at study end (95% CI: 94.1%–97.0%). The study that enrolled patients who were naïve to the ELLIPTA inhaler (200301 [17]) reported the inhaler to be "easy" or "very easy" to use in 92–98% of patients, vs. 93% of patients who correctly used the inhaler in the study that enrolled patients with no use of the ELLIPTA inhaler within the previous 6 months (201071 [18]) (Table 3). Correct use (Fig 3) and user ratings of "easy" or "very easy" (Fig 3) of the placebo ELLIPTA DPI remained high with increasing age, including patients aged ≥65 years. However, among the ≥75 years age group, there was a slight trend towards a decrease in the proportion of patients demonstrating correct use of the placebo ELLIPTA DPI at study end and in the proportion finding its use "very easy" (Figs 3 and 4). The 95% CI for the 40–64 age group's correct use do not overlap with the 65–74 and 75+ year

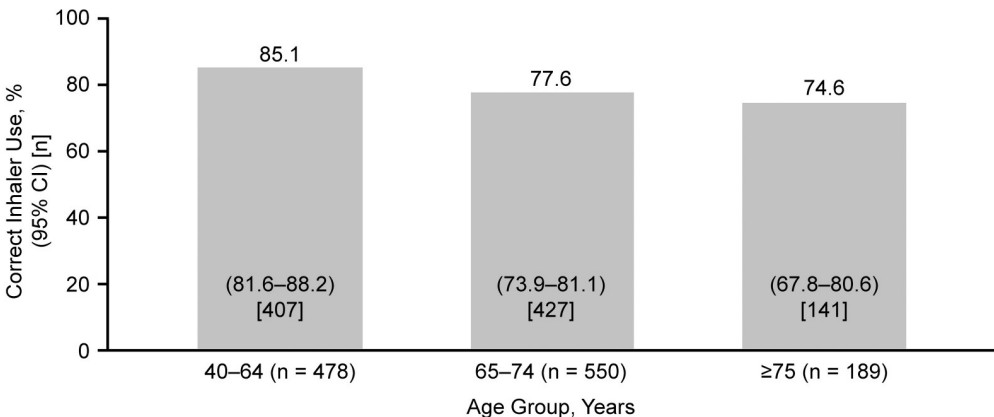

**Fig 3. Correct use of the placebo ELLIPTA DPI at study end by age group (n = 1217).** The 95% CI for the percentages is calculated using the exact binomial distribution. DPI, dry-powder inhaler; CI, confidence interval.

groups, suggesting significant differences–however no statistical modelling was performed to investigate the differences in age groups.

## Safety

Patients continued their ongoing COPD maintenance therapy throughout each of the included studies. The most common on-treatment respiratory COPD medications were salbutamol (31%), budesonide/formoterol fumarate (20%), and tiotropium bromide (19%). All AEs, including treatment-related AEs and SAEs, are summarized in Table 4.

## Discussion

In this analysis, the majority of patients with COPD demonstrated correct use of the placebo ELLIPTA DPI at study end. Of the patients demonstrating correct use who were asked to rate

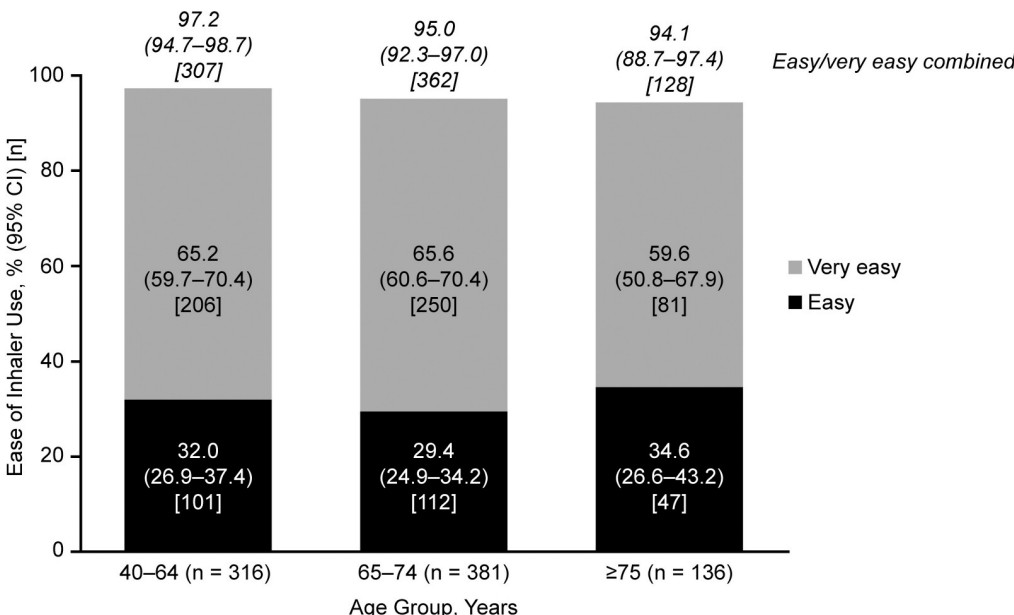

**Fig 4. Ease-of-use of the placebo ELLIPTA DPI at study end by age group (n = 833).** The 95% CI for the percentages is calculated using the exact binomial distribution. CI, confidence interval; DPI, dry-powder inhaler.

**Table 4. Summary of AEs.**

| Event, n (%) | ITT Population (n = 1232) |
|---|---|
| Any AE | 73 (6) |
| AEs that led to study discontinuation | 5 (<1) |
| AEs reported in ≥3 patients | |
| Headache | 10 (<1) |
| Nasopharyngitis | 8 (<1) |
| Back pain | 6 (<1) |
| Sinusitis | 5 (<1) |
| Cough | 5 (<1) |
| COPD | 4 (<1) |
| Arthralgia | 4 (<1) |
| Pneumonia | 3 (<1) |
| Upper respiratory tract infection | 3 (<1) |
| Treatment-related AEs (placebo ELLIPTA DPI only)[a] | |
| Cough | 4 (<1) |
| Back pain | 1 (<1) |
| Dyspnea | 1 (<1) |
| Laceration | 1 (<1) |
| Oral paresthesia | 1 (<1) |
| Any SAE | 5 (<1) |
| SAE related to study treatment | 0 (0) |
| Fatal SAEs | 0 (0) |

[a]One patient had a cough from ELLIPTA powder inhalation; reason for cough was not specified in the other 3 patients. Patients experienced the following AEs secondary to the placebo ELLIPTA DPI: back pain (n = 1), increased dyspnea (n = 1), and oral paresthesia (n = 1). One patient experienced laceration (cuts on both thumbs) while opening the blister card packaging of placebo capsules.

AE, adverse event; COPD, chronic obstructive pulmonary disease; DPI, dry-powder inhaler; ITT, intent-to-treat; SAE, serious adverse event.

inhaler use, almost all rated the inhaler as "easy" or "very easy" to use. Inhaler assessment includes preference, ease-of-use, correct use, critical errors, and training time [22]. The GOLD 2022 report emphasizes the relationship between improper inhaler technique and inadequate symptom control [1], indicating correct use is key as it likely impacts adequate drug delivery to the lungs. Furthermore, improved clinical outcomes are dependent on not only choice of active drug but also choice of inhaler [11]. Therefore, it is important for patients to correctly use their inhaler and find it easy to use; however, this analysis did not investigate the relationship between inhaler use and clinical outcomes.

The results from this analysis are consistent with other clinical [23] and qualitative studies [24] in which many patients with COPD found the placebo ELLIPTA DPI easy to use. Correct placebo ELLIPTA DPI use, and placebo ELLIPTA DPI use that was rated "easy" or "very easy", both remained high in patients aged ≥65 years, suggesting that use of the ELLIPTA DPI would provide appropriate drug delivery to the lungs and would subsequently increase the likelihood of successful clinical outcomes in this patient population. These are encouraging results, as elderly patients with COPD frequently face problems when using inhalers. Up to 94% of patients with COPD make at least 1 error in inhaler technique in clinical settings [2, 25], with the error rate doubling for patients >60 years and quadrupling for patients aged >80 years, compared with patients aged <60 years [11]. COPD prevalence is also seen to increase

with patient age. For example, previous estimates of COPD prevalence among United States (US) adults in 2011 suggested that the prevalence of COPD almost quadrupled from patients aged ≤44 years (3.2%) to those aged ≥65 years (>11.6%) [26], and a literature review of studies in the US, Australia and Europe found that the prevalence of COPD was greatest in patients aged ≥75 years compared with younger ages [27]. It is therefore particularly important that elderly patients are able to use their inhaler correctly and easily.

In a study of responses of patients with COPD for the Global Usability Score questionnaire (developed for assessing and comparing the usability of different inhalation devices simultaneously), the ELLIPTA inhaler was identified as having the highest usability score among tested DPIs (Breezhaler, DISKUS, Genuair, Nexthaler, Spiromax, and Turbohaler). This score was independent of the patient's level of DPI experience. The ELLIPTA inhaler was also rated as the quickest to learn and requiring the shortest time for patients to achieve autonomy [28]. Similarly, in a study comparing first experiences with different inhalers in COPD (pMDI, Aerolizer, HandiHaler, Turbohaler, DISKUS, Breezhaler, ELLIPTA, Easyhaler, DISKHALER and Respimat), most participants were able to use the ELLIPTA inhaler correctly with ≤1 counseling attempts [29].

Few AEs, including treatment-related AEs, were observed which was expected because a placebo version of the ELLIPTA DPI was evaluated and no active drug was administered; therefore, the reported AEs may reflect those related to patients' ongoing COPD maintenance therapy or potentially breakthrough symptoms of underlying disease. The most commonly reported AEs (headache and nasopharyngitis) in this analysis are known side effects of the most common on-treatment respiratory COPD medications used by patients in the included studies (salbutamol [30], budesonide/formoterol fumarate [31], and tiotropium bromide [pharyngitis] [32]).

## Limitations

Only data from the placebo ELLIPTA DPI were included in the pooled analysis, without data on a comparator inhaler for reference. A limitation at outcome level is that the patient population and outcome definitions were not identical across the included studies. Each study, as with all inhaler use studies, faced challenges in defining the key parameters around correct use as no standard definition exists [33]. At study level, a limitation of study 201071 [16] was that screening procedures involved an assessment of correct inhaler use, although few patients were excluded for this reason. Study 201071 [18] allowed recruitment of patients who had not used the ELLIPTA inhaler within 6 months, study 206901 [20] within 12 months and study 206215 [19] within 24 months prior to screening, while study 200301 [17] included only patients who were completely naïve to the inhaler. Therefore, the possibility that the correct ease-of-use data in studies 201071 [18], 206901 [20], and 206215 [19] were influenced by the patients' previous experience with the ELLIPTA inhaler cannot be discounted. However, while there were differences in correct use between each study, perception on ease-of-use appeared to be relatively unaffected (Table 3). Furthermore, there were differences between studies in the number of times patients received correct use training (Table 1), which could have introduced a possible source of inequity in instruction recall and potentially impacted on both correct use and perceived ease-of-use of the inhaler.

The data source was biased; only GSK-sponsored studies were included. Each patient-level dataset was pooled into one main dataset, but no accounts for weighting or clustering of patients were performed. This was considered appropriate due to all 4 studies sharing similar patient characteristics, study designs, and statistical methods. However, the data were not truly homogenous, and the level of heterogeneity was not assessed. For the investigation of age

groups, the 95% confidence intervals between the 40–64 years age group do not overlap with the other two age ranges, suggesting a significant difference in proportions, however no statistical tests comparing the differences between age groups were performed. It should also be noted that the use of unvalidated questionnaires to evaluate the secondary patient-reported outcomes is another limitation of the study. No assessment of bias was calculated. Another potential source of bias was the open-label design of the studies, which relied on subjective assessments by trained HCPs. Existing real-world observational studies have reported that patients continue to misuse inhalers in the real world [8, 9], in contrast to our results that reported combined data from experimental clinical studies. The patients in this analysis may have received more thorough training on correct inhaler technique than in a routine clinical practice setting. Therefore, it is possible that ease-of-use of the placebo ELLIPTA DPI was overestimated, potentially limiting the generalizability of the results to real-world clinical practice. These results further emphasize the importance of HCPs providing patients with proper training in inhaler technique to reduce mishandling and improve drug delivery to the lungs.

## Conclusions

The majority of patients with COPD included in this analysis were able to use the placebo ELLIPTA DPI correctly and found the inhaler "easy" or "very easy" to use. Correct use and ease-of-use findings were consistent across all age groups evaluated, including patients aged ≥65 years. AEs with the placebo ELLIPTA DPI were uncommon. Our findings, in combination with guidance from the GOLD 2022 report, emphasize that proper inhaler training and clear instructions on PILs are important for correct inhaler use and easy handling.

## Supporting information

**S1 Table. Ease-of-use questionnaire (ELLIPTA and non-ELLIPTA) for study 200301 [1].**
(DOCX)

**S2 Table. Ease-of-use questionnaire (ELLIPTA) for study 201071 [1].**
(DOCX)

## Acknowledgments

The authors would like to thank the study participants and staff for their contributions to the original studies included in this analysis. Medical writing support for the development of this manuscript, under the direction of the authors, was provided by Kirsty Millar, MSc, and Andrew Briggs, BA, of Ashfield MedComms (Macclesfield, UK), an Inizio company, and was funded by GSK.

Trademarks are owned by or licensed to their respective owners (the GSK group of companies [ELLIPTA, DISKUS, DISKHALER]; AstraZeneca [Turbuhaler, Genuair]; Boehringer Ingelheim [HandiHaler, Respimat]; Novartis Pharmaceuticals [Breezhaler, Aerolizer], Orion Pharma [Easyhaler], Chiesi [Nexthaler], Teva [Spiromax]).

## Author Contributions

**Conceptualization:** Renu Jain, Kathryn Collison, Raj Sharma, Laura Sutton, Jamie Rees.

**Data curation:** Thomas M. Siler, David I. Bernstein.

**Formal analysis:** Thomas M. Siler, Renu Jain, Kathryn Collison, Raj Sharma, Laura Sutton, Jamie Rees, David I. Bernstein.

**Methodology:** Renu Jain, Kathryn Collison, Raj Sharma, Laura Sutton, Jamie Rees.

**Writing – original draft:** Thomas M. Siler, Renu Jain, Kathryn Collison, Raj Sharma, Laura Sutton, Jamie Rees, David I. Bernstein.

**Writing – review & editing:** Thomas M. Siler, Renu Jain, Kathryn Collison, Raj Sharma, Laura Sutton, Jamie Rees, David I. Bernstein.

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
