## [Decision Letter · Decision Letter 0]

9 Feb 2022

PONE-D-21-22827Correct use and ease-of-use of placebo ELLIPTA dry-powder inhaler in adult patients with chronic obstructive pulmonary diseasePLOS ONE

Dear Dr. Renu Jain,

Thank you for submitting your manuscript to PLOS ONE. After careful consideration, we feel that it has merit but does not fully meet PLOS ONE’s publication criteria as it currently stands. Therefore, we invite you to submit a revised version of the manuscript that addresses the points raised during the review process.

Please answer point by point to reviewers to increase the quality of the manuscript, of interest and well written.

We look forward to receiving your revised manuscript.

Kind regards,

Manlio Milanese

Academic Editor

PLOS ONE

Journal Requirements:

This work was funded by GlaxoSmithKline plc. (study 212146).

3. Thank you for stating the following in the Acknowledgments/ Funding Section of your manuscript: 

This work was funded by GlaxoSmithKline plc. (study 212146).

This work was funded by GlaxoSmithKline plc. (study 212146).

I have read the journal’s policy and the author of this manuscript have the following competing interests: RJ, KC, RS, LS, and JR report employment with, and stock/share ownership in, GlaxoSmithKline plc. during study conduct. KC is currently employed by AstraZeneca and LS is no longer employed by GlaxoSmithKline plc. TMS received research support from West-Ward Pharmaceuticals, Theravance Biopharma US, Inc., GlaxoSmithKline plc., Pearl Therapeutics, Chiesi, AstraZeneca, Novartis, Boehringer Ingelheim, Forest, Compleware, Evidera, Oncocyte, Teva, Vapotherm, Sunovion, Proterix BioPharma, Seer, and Sanofi. TMS has also received speaker fees from GlaxoSmithKline plc., Mylan Inc./Theravance Biopharma US, Inc., and Sunovion, and consulting fees from Vapotherm. DIB received grant/research/clinical trial support from GlaxoSmithKline plc., Teva, AstraZeneca, Pearl Therapeutics, Novartis, Genentech, Inc., Merck, Boehringer Ingelheim, Amgen, Aimmune, Shire, and Biocryst and consulted/participated in advisory boards for GlaxoSmithKline plc., ALK America, Gerson-Lehman, and Guidepoint Global.

Additional Editor Comments:

This retrospective study on pooled data coming from 4 studies assessing use of Ellipta DPI in COPD patients is of clinical interest and well-written. Please, answer point by point to the two reviewers to increase the quality of the manuscript.

Reviewers' comments:

Reviewer's Responses to Questions

**Comments to the Author**

1. Is the manuscript technically sound, and do the data support the conclusions?

Reviewer #1: Yes

Reviewer #2: Yes

2. Has the statistical analysis been performed appropriately and rigorously? 

Reviewer #1: Yes

Reviewer #2: Yes

3. Have the authors made all data underlying the findings in their manuscript fully available?

Reviewer #1: Yes

Reviewer #2: Yes

4. Is the manuscript presented in an intelligible fashion and written in standard English?

Reviewer #1: Yes

Reviewer #2: Yes

5. Review Comments to the Author

Reviewer #1: This is a retrospective study on pooled data coming from 4 studies assessing use of Ellipta DPI in COPD patients. The paper is of interest for the readers of the Journal and results sound.

I have the following question that the Authors need to address. Scrutiny of the figure 3 seems to indicate that there is a significant (and not a trend as reported in the text) difference in the % of patients using Ellipta correctly between patients aged 40-64 and those older than 75 yr (see CIs bewteen groups). Please check it with appropriate statiscal methods.

Reviewer #2: I would like to congratulate the authors for this interesting and well-written work. I have just

a few comments that should be addressed in the manuscript.

- The secondary endpoint was the percentage of patients rating the placebo

ELLIPTA DPI as “easy” or “very easy” to use at study end, as assessed by

questionnaires in patients deemed to be using the placebo ELLIPTA DPI correctly at

study end. What does it mean? Did the Authors consider only question 1 of the questionnaires?

- The secondary outcome has been assessed by mean of unvalidated tools. This is something that should be discussed. The use of unvalidated questionnaires to evaliutate the patient reported outcomes is a limit.

6. PLOS authors have the option to publish the peer review history of their article (what does this mean?). If published, this will include your full peer review and any attached files.

Reviewer #1: **Yes: **Federico Lavorini

Reviewer #2: **Yes: **Ilaria Baiardini

---

## [Author Response · Author response to Decision Letter 0]

18 May 2022

Renu Jain

GlaxoSmithKline plc.

Research Triangle Park

Durham

North Carolina

United States

Manlio Milanese

31 March 2022

Dear Dr Milanese,

On behalf of my co-authors, I would like to thank you for considering our manuscript for publication and providing us with the opportunity to address comments and make minor revisions suggested by the Editorial Team and Reviewers. Please find clean and tracked changes versions of our revised manuscript enclosed, and our point-by-point replies to the comments below in blue italicised font, where additions have been made bold and deletions have been crossed out. Please note that page and line numbers refer to the version of the manuscript with tracked changes. I hope you will now consider our article suitable for publication in PLOS ONE. Please do let me know if anything further is required. We look forward to receiving your decision.

Yours sincerely,

Renu Jain

GlaxoSmithKline plc.

 

Journal Requirements: 

We have reviewed the templates and can confirm our manuscript adheres to PLOS ONE’s style requirements

This work was funded by GlaxoSmithKline plc. (study 212146).

We have removed the funding statement from the manuscript (page 25, lines 370–371). Please include the following statement in the online submission form:

“This work was funded by GlaxoSmithKline plc. (Study 212146). The funders had a role in study design, data collection and analysis, decision to publish, and preparation of the manuscript.”

3. Thank you for stating the following in the Acknowledgments/ Funding Section of your manuscript: 

This work was funded by GlaxoSmithKline plc. (study 212146).

This work was funded by GlaxoSmithKline plc. (study 212146).

Thank you for highlighting this. We have removed the funding statement from the manuscript (page 25, lines 370–371) and have included the amended wording above in Comment 2.

I have read the journal’s policy and the author of this manuscript have the following competing interests: RJ, KC, RS, LS, and JR report employment with, and stock/share ownership in, GlaxoSmithKline plc. during study conduct. KC is currently employed by AstraZeneca and LS is no longer employed by GlaxoSmithKline plc. TMS received research support from West-Ward Pharmaceuticals, Theravance Biopharma US, Inc., GlaxoSmithKline plc., Pearl Therapeutics, Chiesi, AstraZeneca, Novartis, Boehringer Ingelheim, Forest, Compleware, Evidera, Oncocyte, Teva, Vapotherm, Sunovion, Proterix BioPharma, Seer, and Sanofi. TMS has also received speaker fees from GlaxoSmithKline plc., Mylan Inc./Theravance Biopharma US, Inc., and Sunovion, and consulting fees from Vapotherm. DIB received grant/research/clinical trial support from GlaxoSmithKline plc., Teva, AstraZeneca, Pearl Therapeutics, Novartis, Genentech, Inc., Merck, Boehringer Ingelheim, Amgen, Aimmune, Shire, and Biocryst and consulted/participated in advisory boards for GlaxoSmithKline plc., ALK America, Gerson-Lehman, and Guidepoint Global.

We have included the following in the revised manuscript:

Author disclosure statements, page 25, lines 355–368: “RJ, KC, RS, LS, and JR report employment with, and stock/share ownership in, GlaxoSmithKline plc. during study conduct. KC is currently employed by AstraZeneca and LS is no longer employed by GlaxoSmithKline plc. TMS received research support from West-Ward Pharmaceuticals, Theravance Biopharma US, Inc., GlaxoSmithKline plc., Pearl Therapeutics, Chiesi, AstraZeneca, Novartis, Boehringer Ingelheim, Forest, Compleware, Evidera, Oncocyte, Teva, Vapotherm, Sunovion, Proterix BioPharma, Seer, and Sanofi. TMS has also received speaker fees from GlaxoSmithKline plc., Mylan Inc./Theravance Biopharma US, Inc., and Sunovion, and consulting fees from Vapotherm. DIB received grant/research/clinical trial support from GlaxoSmithKline plc., Teva, AstraZeneca, Pearl Therapeutics, Novartis, Genentech, Inc., Merck, Boehringer Ingelheim, Amgen, Aimmune, Shire, and Biocryst and consulted/participated in advisory boards for GlaxoSmithKline plc., ALK America, Gerson-Lehman, and Guidepoint Global. This does not alter our adherence to PLOS ONE policies on sharing data and materials.”

We have updated the Data Availability statement as follows:

Data Availability statement, page 24, lines 348–351: “Anonymized individual participant data and study documents can be requested for further research from www.clinicalstudydatarequest.com.

The authors confirm that the data supporting the findings of this study are available within the article and its supplementary materials.”

Thank you for this request. We have added the following ORCID details to the title page of the manuscript:

Title page, page 1, lines 19–21: “ORCID details

Thomas M. Siler 0000-0002-5103-0360

Renu Jain 0000-0001-5400-2536”

Please may we kindly advise that the YouTube link in the above comment is linked to a private video, so the content cannot be accessed.

We have reviewed the reference list and can confirm it is complete, up to date and correct, with no retracted references.

Additional Editor Comments:

This retrospective study on pooled data coming from 4 studies assessing use of Ellipta DPI in COPD patients is of clinical interest and well-written. Please, answer point by point to the two reviewers to increase the quality of the manuscript.

 

Reviewers' Comments to the Author:

Reviewer #1

This is a retrospective study on pooled data coming from 4 studies assessing use of Ellipta DPI in COPD patients. The paper is of interest for the readers of the Journal and results sound. I have the following question that the Authors need to address.

We thank the Reviewer for their kind words about our manuscript.

Scrutiny of the figure 3 seems to indicate that there is a significant (and not a trend as reported in the text) difference in the % of patients using Ellipta correctly between patients aged 40-64 and those older than 75 years (see CIs between groups). Please check it with appropriate statistical methods.

Thank you for this query. As there were no statistical analyses performed on the data, it is not possible to state if the results were significant. We have added the following to the results and discussion sections as follows:

Results, page 14, lines 184–192: “Correct use (Fig 3) and user ratings of “easy” or “very easy” (Fig 3) of the placebo ELLIPTA DPI remained high with increasing age, including patients aged ≥65 years. However, among the ≥75 years age group, there was a slight trend towards a decrease in the proportion of patients demonstrating correct use of the placebo ELLIPTA DPI at study end and in the proportion finding its use “very easy” (Fig 3, 4). The 95% CI for the 40-64 age group’s correct use do not overlap with the 

65–74 and 75+ year groups, suggesting significant differences – however no statistical modelling was performed to investigate the differences in age groups.”

Discussion, page 22, lines 302–306: “However, the data were not truly homogenous, and the level of heterogeneity was not assessed. For the investigation of age groups, the 95% confidence intervals between the 40–64 years age group do not overlap with the other two age ranges, suggesting a significant difference in proportions, however no statistical tests comparing the differences between age groups were performed.”

Reviewer #2

I would like to congratulate the authors for this interesting and well-written work. I have just a few comments that should be addressed in the manuscript.

We thank the Reviewer for their kind words about our manuscript.

The secondary endpoint was the percentage of patients rating the placebo

ELLIPTA DPI as “easy” or “very easy” to use at study end, as assessed by

questionnaires in patients deemed to be using the placebo ELLIPTA DPI correctly at

study end. What does it mean? Did the Authors consider only question 1 of the questionnaires?

Thank you for this query. We can confirm that only question 1 of the questionnaires was considered as part of this analysis, and further analyses might consider the remaining questions that were posed (ability to tell how many doses remained, likelihood of asking for ELLIPTA if their medications were available in the inhaler). We believe that the meaning of this result is that patients find it easy to use the inhaler, regardless of prior experience with it. This result is supported by the literature, and the meaning of this result and its context in the wider literature is included in the discussion section as follows: 

Discussion, page 19, lines 241–245: “Correct placebo ELLIPTA DPI use, and placebo ELLIPTA DPI use that was rated “easy” or “very easy”, both remained high in patients aged ≥65 years, suggesting that use of the ELLIPTA DPI would provide appropriate drug delivery to the lungs and would subsequently increase the likelihood of successful clinical outcomes in this patient population.”

Discussion, page 20, lines 257–267: “In a study of responses of patients with COPD for the Global Usability Score questionnaire (developed for assessing and comparing the usability of different inhalation devices simultaneously), the ELLIPTA inhaler was identified as having the highest usability score among tested DPIs (Breezhaler, DISKUS, Genuair, Nexthaler, Spiromax, and Turbohaler). This score was independent of the patient’s level of DPI experience. The ELLIPTA inhaler was also rated as the quickest to learn and requiring the shortest time for patients to achieve autonomy [28]. Similarly, in a study comparing first experiences with different inhalers in COPD (pMDI, Aerolizer, HandiHaler, Turbohaler, DISKUS, Breezhaler, ELLIPTA, Easyhaler, DISKHALER and Respimat), most participants were able to use the ELLIPTA inhaler correctly with ≤1 counseling attempts [29].”

However, there were several limitations in the study designs regarding the use of the ELLIPTA inhaler which are noted in the discussion section:

Discussion, page 21, lines 283–291: “At study level, a limitation of study 201071 [16] was that screening procedures involved an assessment of correct inhaler use, although few patients were excluded for this reason. Study 201071 [18] allowed recruitment of patients who had not used the ELLIPTA inhaler within 6 months, study 206901 [20] within 12 months and study 206215 [19] within 24 months prior to screening, while study 200301 [17] included only patients who were completely naïve to the inhaler. Therefore, the possibility that the correct ease-of-use data in studies 201071 [18], 206901 [20], and 206215 [19] were influenced by the patients’ previous experience with the ELLIPTA inhaler cannot be discounted.”

The secondary outcome has been assessed by mean of unvalidated tools. This is something that should be discussed. The use of unvalidated questionnaires to evaluate the patient reported outcomes is a limit.

We thank the Reviewer for highlighting this to us. We have added the following to the discussion section:

Discussion, page 22, lines 302–308: “However, the data were not truly homogenous, and the level of heterogeneity was not assessed. For the investigation of age groups, the 95% confidence intervals between the 40–64 years age group do not overlap with the other two age ranges, suggesting a significant difference in proportions, however no statistical tests comparing the differences between age groups were performed. It should also be noted that the use of unvalidated questionnaires to evaluate the secondary patient-reported outcomes is another limitation of the study.”

---

## [Decision Letter · Decision Letter 1]

4 Aug 2022

Correct use and ease-of-use of placebo ELLIPTA dry-powder inhaler in adult patients with chronic obstructive pulmonary disease

PONE-D-21-22827R1

Dear Dr. Renu Jain,

We’re pleased to inform you that your manuscript has been judged scientifically suitable for publication and will be formally accepted for publication once it meets all outstanding technical requirements.

Kind regards,

Manlio Milanese

Academic Editor

PLOS ONE

Additional Editor Comments (optional):

The article, in its revised version, is suotable for publication

Reviewers' comments:

Reviewer's Responses to Questions

**Comments to the Author**

1. If the authors have adequately addressed your comments raised in a previous round of review and you feel that this manuscript is now acceptable for publication, you may indicate that here to bypass the “Comments to the Author” section, enter your conflict of interest statement in the “Confidential to Editor” section, and submit your "Accept" recommendation.

Reviewer #2: All comments have been addressed

2. Is the manuscript technically sound, and do the data support the conclusions?

Reviewer #2: Yes

3. Has the statistical analysis been performed appropriately and rigorously? 

Reviewer #2: Yes

4. Have the authors made all data underlying the findings in their manuscript fully available?

Reviewer #2: Yes

5. Is the manuscript presented in an intelligible fashion and written in standard English?

Reviewer #2: Yes

6. Review Comments to the Author

Reviewer #2: (No Response)

7. PLOS authors have the option to publish the peer review history of their article (what does this mean?). If published, this will include your full peer review and any attached files.

Reviewer #2: **Yes: **Ilaria Baiardini

---

## [Editor Report · Acceptance letter]

5 Aug 2022

PONE-D-21-22827R1 

Correct use and ease-of-use of placebo ELLIPTA dry-powder inhaler in adult patients with chronic obstructive pulmonary disease 

Dear Dr. Jain:

I'm pleased to inform you that your manuscript has been deemed suitable for publication in PLOS ONE. Congratulations! Your manuscript is now with our production department. 

Kind regards, 

on behalf of

Dr. Manlio Milanese 

Academic Editor

PLOS ONE